 **eLIFE** elifesciences.org

# Clathrin coat controls synaptic vesicle acidification by blocking vacuolar ATPase activity

**Zohreh Farsi[1,2], Sindhuja Gowrisankaran[3†], Matija Krunic[3†], Burkhard Rammner[4], Andrew Woehler[2], Eileen M Lafer[5,6], Carsten Mim[7,8], Reinhard Jahn[1], Ira Milosevic[3]***

[1]Department of Neurobiology, Max Planck Institute for Biophysical Chemistry, Göttingen, Germany; [2]Berlin Institute for Medical Systems Biology, Max Delbrück Center for Molecular Medicine, Berlin, Germany; [3]Synaptic Vesicle Dynamics Group, European Neuroscience Institute, University Medical Center Göttingen, Göttingen, Germany; [4]Sciloop, Hamburg, Germany; [5]Department of Biochemistry and Structural Biology, University of Texas Health Science Center at San Antonio, San Antonio, United States; [6]Center for Biomedical Neuroscience, University of Texas Health Science Center at San Antonio, San Antonio, United States; [7]Department for Biomedical Engineering and Health Solutions, Kungliga Tekniska Högskolan, Huddinge, Sweden; [8]Department of Biosciences and Nutrition, Karolinska Institute, Huddinge, Sweden

**Abstract** Newly-formed synaptic vesicles (SVs) are rapidly acidified by vacuolar adenosine triphosphatases (vATPases), generating a proton electrochemical gradient that drives neurotransmitter loading. Clathrin-mediated endocytosis is needed for the formation of new SVs, yet it is unclear when endocytosed vesicles acidify and refill at the synapse. Here, we isolated clathrin-coated vesicles (CCVs) from mouse brain to measure their acidification directly at the single vesicle level. We observed that the ATP-induced acidification of CCVs was strikingly reduced in comparison to SVs. Remarkably, when the coat was removed from CCVs, uncoated vesicles regained ATP-dependent acidification, demonstrating that CCVs contain the functional vATPase, yet its function is inhibited by the clathrin coat. Considering the known structures of the vATPase and clathrin coat, we propose a model in which the formation of the coat surrounds the vATPase and blocks its activity. Such inhibition is likely fundamental for the proper timing of SV refilling.
DOI: https://doi.org/10.7554/eLife.32569.001

*For correspondence:
imilose@gwdg.de

†These authors contributed equally to this work

## Introduction

Neuronal synapses are capable of regenerating SVs locally with high efficiency and fidelity in order to meet the demands of neuronal activity. The uniquely homogeneous sizes of SVs and their defined protein composition suggest the existence of a precise endocytic machinery that shapes and promotes the fission of SVs, either from the plasma membrane and/or endosome-like structures (*Takamori et al., 2006*; *Saheki and De Camilli, 2012*; *Soykan et al., 2016*; *Milosevic, 2018a*). Clathrin-mediated endocytosis is a classic example of vesicle formation mediated by a coat assembly, and it occurs at the synapse (*Saheki and De Camilli, 2012*; *Milosevic, 2018a*). It has been intensely studied for over four decades, yet numerous details, including when the endocytosed vesicle initiates acidification, remain unclear. The timing and regulation of vesicle acidification is an essential question, not only in the context of SV recycling and neurotransmitter uptake, but also for other

endocytic pathways where lowering of the luminal pH is vital for a range of functions, such as separation of cargo from receptor (e.g. iron from transferrin; *Grant and Donaldson, 2009*) or the activation of viral fusion proteins (*White et al., 2008*).

Acidification of endocytic compartments is primarily mediated by vATPases. These proton pumps are large complexes composed of 14 different subunits that are organized into an ATP-hydrolytic domain ($V_1$) and a proton-translocation domain ($V_o$) (*Imamura et al., 2003*; *Marshansky et al., 2014*; *Cotter et al., 2015*; *Mazhab-Jafari et al., 2016*). vATPases are particularly well studied at the synapse, where they traffic with other SV proteins through the SV cycle and generate a proton electrochemical gradient ($\Delta\mu_{H+}$) across the vesicular membrane, fueling the reloading of SVs with neurotransmitters. Yet, many details are not well understood: is vATPase fully active during the entire SV cycle, or is it regulated? If it is regulated, how and when is that done? While some have suggested that endocytic coated vesicles do not have acidic internal pH (*Anderson et al., 1984*; *Anderson and Orci, 1988*), other studies have shown that membrane-permeable, pH-sensitive, weak bases accumulate in the lumen of CCVs, indicating that acidification occurs in the presence of the coat (*Forgac et al., 1983*; *Van Dyke et al., 1984*; *Van Dyke et al., 1985*). To weigh in on this debate, and to investigate whether vATPase is active on CCVs, we have performed a full characterization of the $\Delta\mu_{H+}$ at the single CCV level.

## Results

We have prepared CCVs from mouse brain by adapting a published protocol (*Maycox et al., 1992*) (*Figure 1A*, *Figure 1—figure supplement 1A*). Negative-stained electron microscopy (EM) images showed that the CCVs were abundant while almost no uncoated structures were present (*Figure 1B*, *Figure 1—figure supplement 1B*). Immunoblotting for organellar marker proteins revealed high enrichment of synaptic vesicle proteins as well as clathrin and clathrin adaptors, whereas no significant presence of plasma membrane, endosome and proteasomal proteins was detected (*Figure 1C*). Detailed analysis of CCV samples by mass spectrometry revealed that the large majority of CCVs seems to be derived from the synapses, yet some CCV come from neuronal cell body/intracellular membranes (according to the AP2/AP1 ratio; *Supplementary file 1*). Furthermore, cryo-EM classification of 43,711 individual particles revealed that the majority of vesicles had an intact clathrin coat (*Figure 1—figure supplement 2A*). The reconstruction from major classes of the imaged particles resulted in a highly symmetric structure (*Figure 1D*; *Figure 1—figure supplement 2B–E*) (*Video 1*), which was comparable in size and symmetry to reconstructed 'barrel-like' empty clathrin cages (*Figure 1—figure supplement 2F*) (*Fotin et al., 2004*). The median size of CCVs (detailed in *Figure 1D* for the reconstructed CCV structure and in *Figure 1—figure supplement 1C* for the population of CCVs) is in line with previous measurements of CCVs from pig and bovine brains (*Pearse, 1975*; *Nossal et al., 1983*).

The $\Delta\mu_{H+}$ is composed of a chemical ($\Delta pH$) and an electrical gradient ($\Delta\psi$) which together provide the free energy required for loading the neurotransmitters in the vesicular lumen. Both $\Delta\mu_{H+}$ components are important for the efficient import of different neurotransmitters, and can be regulated differently in various organelles (*Farsi et al., 2017*). To fully characterize the $\Delta\mu_{H+}$ in CCVs, we performed measurements of $\Delta pH$ and $\Delta\psi$ at a single vesicle level, as described earlier (*Farsi et al., 2016*). $\Delta pH$ was measured in CCVs isolated from brains of mice expressing synaptopHluorin (spH, super-ecliptic pHluorin tagged to the luminal domain of VAMP2; *Li et al., 2005*) in the vesicular lumen (spH-CCVs), while $\Delta\psi$ was measured in CCVs isolated from the wild-type mouse brains after labeling with the potentiometric probe VF2.1.Cl (*Miller et al., 2012*). The isolated CCVs were immobilized on glass coverslips, imaged under total internal reflection fluorescence (TIRF) illumination, and their fluorescence in response to ATP was measured (*Figure 2A–B*; *Figure 2—figure supplement 1*). The same measurements, as well as the basic characterization of vesicle acidification properties (*Figure 2—figure supplement 2*), were performed with SVs isolated from the brains of transgenic mice (spH-SVs) and wild-type mice after labeling with VF2.1.Cl. Upon addition of ATP, only a minor fluorescence change was detected in CCVs when compared to the respective signals from the SVs (*Figure 2C–D*), indicating that the ATP-induced change in pH and membrane potential was much smaller in CCVs (*Figure 2E–F*). The distribution of fluorescence response of single CCVs and SVs clearly showed that the majority of the CCVs did not display any ATP-induced acidification (*Figure 2G–H*). The same results were obtained in the presence of chloride in the glycine buffer

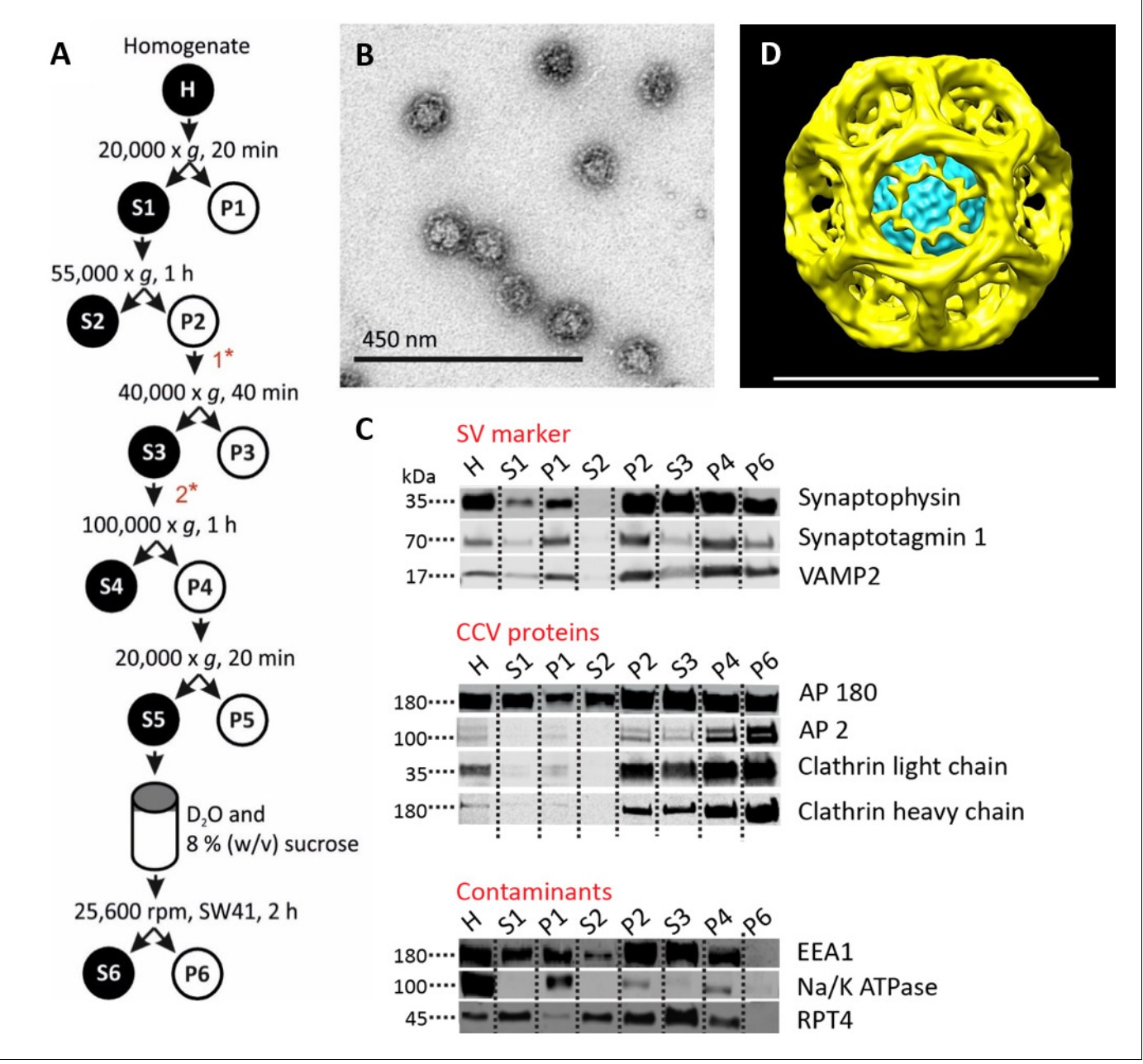

**Figure 1.** Isolation of clathrin-coated vesicles from mouse brain. (**A**) Schema illustrating the isolation procedure of CCVs from mouse brains. Numbers 1 and 2 represent steps where Ficol and sucrose (at final conc. of 6.25 (wt/v)) were added to the sample, and where the sample was diluted 5x with buffer, respectively. See Suppl. Data for details. (**B**) Electron micrograph of isolated CCVs after negative staining. (**C**) Immunoblots of fractions collected during the CCV isolation protocol for various marker proteins (proteins were separated at the 4–15% gradient gel and detected by the Li-COR Odyssey imaging system). See also *Supplementary file 1* for mass spectrometry analysis of SVs and CCVs samples. (**D**) Reconstructed CCV from 6114 particles sized up to a diameter of 80 nm, with a D6 symmetry imposed, shows that ex vivo CCVs adopt the same 'barrel-like' structure as previously reported, and reveal the position of vesicle in its center (see also Suppl. Data and *Figure 1—figure supplement 2*). The median values extracted from the reconstruction of clathrin coat and vesicle within the coat are (note that the structure is a barrel): coated vesicle diameter 75 nm x 73.5 nm, coat thickness 15 nm, vesicle diameter 40 × 35 nm.
DOI: https://doi.org/10.7554/eLife.32569.002

The following figure supplements are available for figure 1:

**Figure supplement 1.** Optimization of a procedure for the CCV isolation from mouse brains, and characterization of CCV size.
DOI: https://doi.org/10.7554/eLife.32569.003

**Figure supplement 2.** Determining the quality and the properties of clathrin coat of CCVs isolated from mouse brains by cryo-EM.

*Figure 1 continued on next page*

*Figure 1 continued*

DOI: https://doi.org/10.7554/eLife.32569.004

(*Figure 2—figure supplement 3*). The histogram of luminal pH of CCVs after ATP addition was fit to two Gaussian models revealing that a subpopulation composed of ~6% of the total CCVs acidified to the same mean extent as SVs (*Figure 2I–K*). Acidification of this small subpopulation is likely due to the presence of small SV contamination and/or damaged clathrin coats, as revealed by EM data (*Figure 2—figure supplement 4*). Altogether, our data provide evidence that a significantly smaller $\Delta\mu_{H+}$ is formed across the vesicular membrane in the presence of a clathrin coat. This finding complements previous studies where delayed kinetics of spH quenching was observed during post-stimulus recovery at living synapses without synaptojanin-1, auxilin or endophilin-A that show delayed uncoating and prominently accumulate CCVs (*Mani et al., 2007*; *Yim et al., 2010*; *Milosevic et al., 2011*).

The disruption of acidification in CCVs is likely not due to the absence of a vATPase complex on these vesicles (*Forgac et al., 1983*; *Stone et al., 1983*). To verify this, we performed mass-spectrometry and immunoblotting for vATPase subunits in both CCV and SV samples (*Figure 3A*; *Figure 3—figure supplement 1*; *Supplementary file 1*). Indeed, both $V_o$ and $V_1$ complex subunits were present on CCVs as well as on SVs. However, due to the abundance of coat proteins in the CCV samples, we detected lower levels of vATPase subunits as well as other SV proteins in the CCV sample (total protein levels were equal in both samples; *Figure 3B*). The $V_o/V_1$ ratio was close to one in both samples, showing that a complete vATPase complex was present on SVs and CCVs (*Figure 3C*). Next, in order to test whether these vATPase complexes are active and able to hydrolyze ATP, we measured the ATPase activity in both CCVs and SVs (*Figure 3—figure supplement 2*), and observed that CCVs show significantly less ATPase activity compared to the same amount of SVs (*Figure 3D*). In addition, the ATPase activity in CCVs was not blocked by N-ethylmaleimide (NEM), an inhibitor of vATPase, showing that the remaining ATPase activity in CCVs was not due to the vATPase (*Figure 3E*). These data show that the lower ATPase activity measured in CCVs was likely due to impaired function of vATPases in CCVs. Taken together, vATPases are present on CCVs, but are not able to hydrolyze ATP and pump protons, resulting in inhibited $\Delta\mu_{H+}$ generation.

The most plausible hypothesis to explain these results is that there are intact and functional vATPases on CCVs that are inhibited by the clathrin coat. In this scenario, vATPases should regain their function once the coat is removed. To test this hypothesis, we performed an in vitro uncoating by treating the CCVs with 300 mM Tris-buffer pH 9.0 (as in *Maycox et al., 1992*), followed by the measurements of $\Delta$pH and $\Delta\psi$ in uncoated vesicles in the buffer with the neutral pH, as described above. Firstly, we checked for the uncoating efficiency after alkaline Tris-buffer treatment with negative-staining EM, and observed that almost all vesicles were uncoated (*Figure 4A*). The immunoblot analysis against clathrin heavy chain (HC) and light chain (LC) as well as adaptor proteins AP180 and AP2 on the uncoated vesicles and in the supernatant after uncoating has revealed that almost no clathrin HC, clathrin LC AP2 and AP180 proteins were detected on uncoated vesicles (all proteins were detected in the supernatant, *Figure 4B*), providing further evidence for a complete loss of clathrin lattice. When $\Delta$pH and $\Delta\psi$ in uncoated vesicles was determined, we observed, intriguingly, that uncoated vesicles reached the same luminal pH and membrane potential as SVs upon application of ATP (*Figure 4C–D*). As a control, we performed the same alkaline Tris-buffer treatment with SVs, and observed no difference in the magnitude of $\Delta\mu_{H+}$ in Tris-treated SVs, ruling out any perturbation caused by alkaline Tris-buffer to the vATPase function (*Figure 4C–D*). In addition to the uncoating of CCVs as detailed above, we have used other approaches to uncoat CCVs, namely by treating CCVs with 500 mM Tris pH 7.0, 5 mM DTT, 1 mM PMSF at 4°C for 4 hr (*Prasad and Keen, 1991*), or by adding purified auxilin, Hsp70 and Hsp110 proteins (to mimic

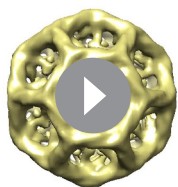

**Video 1.** Reconstructed clathrin-coated vesicle from 6114 raw cryo-EM images of coated vesicle sized up to 80 nm, with D6 symmetry imposed.

DOI: https://doi.org/10.7554/eLife.32569.005

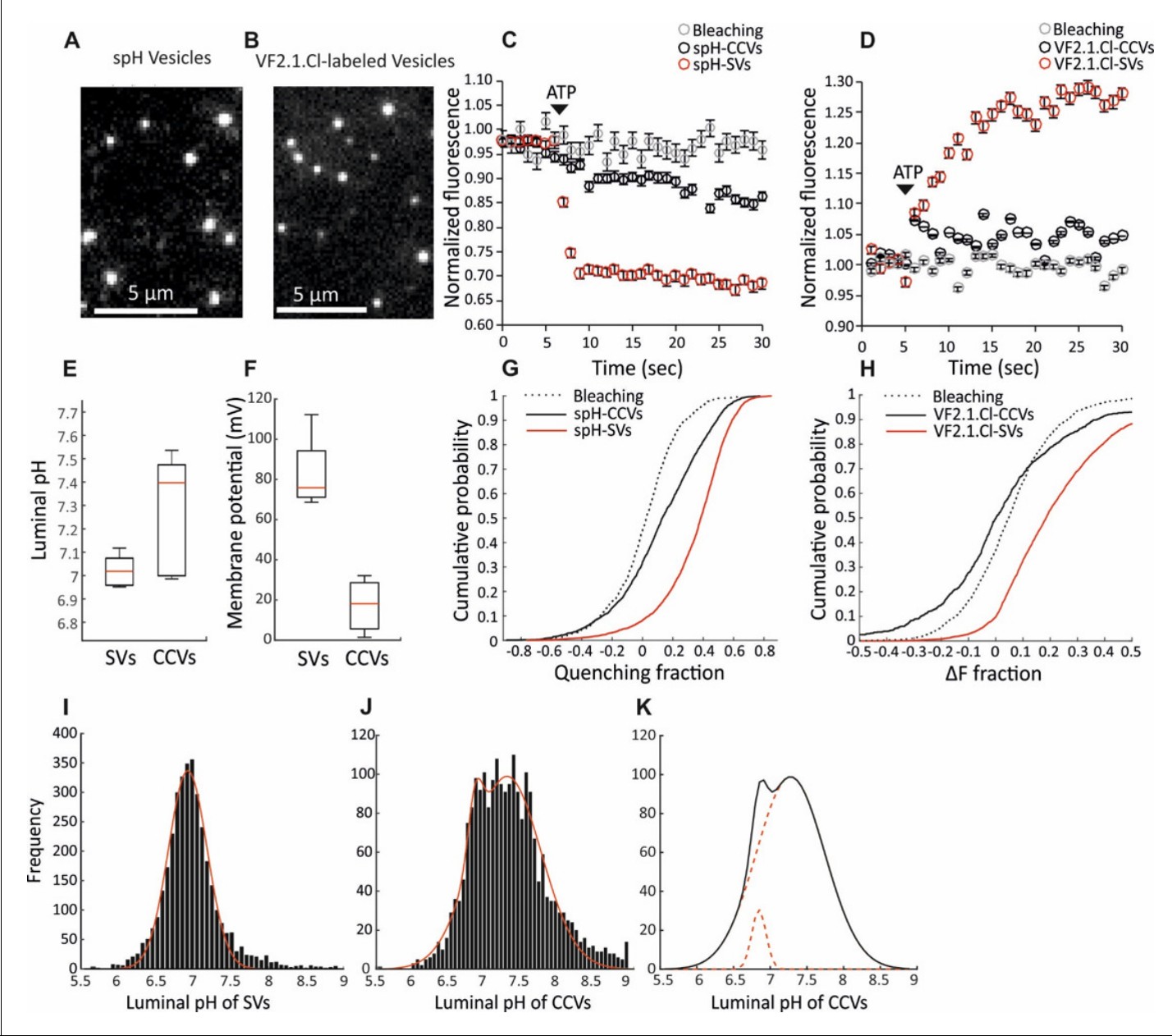

**Figure 2.** Measurement of the electrochemical gradient in SVs and CCVs. (**A–B**) Representative images of single (**A**) spH-CCVs and (**B**) VF2.1.Cl-labeled CCVs using TIRF microscopy. (**C–D**) Averaged fluorescence traces of single (**C**) spH-SV and spH-CCVs, and (**D**) VF2.1.Cl-labeled SVs and VF2.1.Cl-labeled CCVs over time in response to ATP. The control traces indicate the fluorescence response of the spH-CCVs or VF2.1.Cl-labeled CCVs over experimental timescale without ATP addition (the same traces were obtained for spH-SVs and VF2.1.Cl-labeled SVs). Error bars indicate SD from more than 1000 vesicles compiled from 4 to 7 experimental replicates. (**E**) Box plot representation of luminal pH of single SVs and CCVs after addition of 1 mM ATP (box: 1 st and 3rd quartile, line: median, whiskers: the minimum and maximum values). Note that the luminal pH of vesicles equilibrates to 7.4 as shown in *Farsi et al. (2016)*. (**F**) Box plot representation of membrane potential of single SVs and CCVs after addition of 3 mM ATP. (**G–H**) Cumulative frequency plots generated from fluorescence change associated with ATP addition in (**G**) spH-SVs and spH-CCVs, and (**H**) VF2.1.Cl-labeled SVs and VF2.1.Cl-labeled CCVs. The dotted line indicates the fluorescence response of the probes over experimental timescale without ATP addition. (**I–J**) Histograms representing the luminal pH of (**I**) spH-SVs (n = 3,625) and (**J**) spH-CCVs (n = 2,233) upon addition of 1 mM ATP. Red lines indicate single and two-Gaussian models to SV and CCV populations, respectively. (**K**) The population of vesicles contributing to the lower pH (likely CCVs with damaged coat and/or very few SVs) in the CCV population consists of about 6% of the total vesicles measured.
DOI: https://doi.org/10.7554/eLife.32569.006

The following figure supplements are available for figure 2:

**Figure supplement 1.** Flow chart of the single vesicle assay for measuring pH and membrane potential of SVs and CCVs.

*Figure 2 continued on next page*

*Figure 2 continued*

DOI: https://doi.org/10.7554/eLife.32569.007

**Figure supplement 2.** Full acidification of spH-SVs to low luminal pH.

DOI: https://doi.org/10.7554/eLife.32569.008

**Figure supplement 3.** Impairment of acidification in CCVs in the presence of chloride.

DOI: https://doi.org/10.7554/eLife.32569.009

**Figure supplement 4.** EM image of a CCV with a damaged coat.

DOI: https://doi.org/10.7554/eLife.32569.010

physiological conditions) for 15 min at room temperature (*Schuermann et al., 2008*). Both treatments have successfully uncoated all CCVs as verified by EM (*Figure 4—figure supplement 1*), and we have detected full acidification of uncoated vesicles (*Figure 4E*). Taken together, these experiments demonstrate unequivocally that the vATPase activity is inhibited in the presence of a fully assembled clathrin coat. This inhibition is reversible and the vATPase regains its function once the coat is removed.

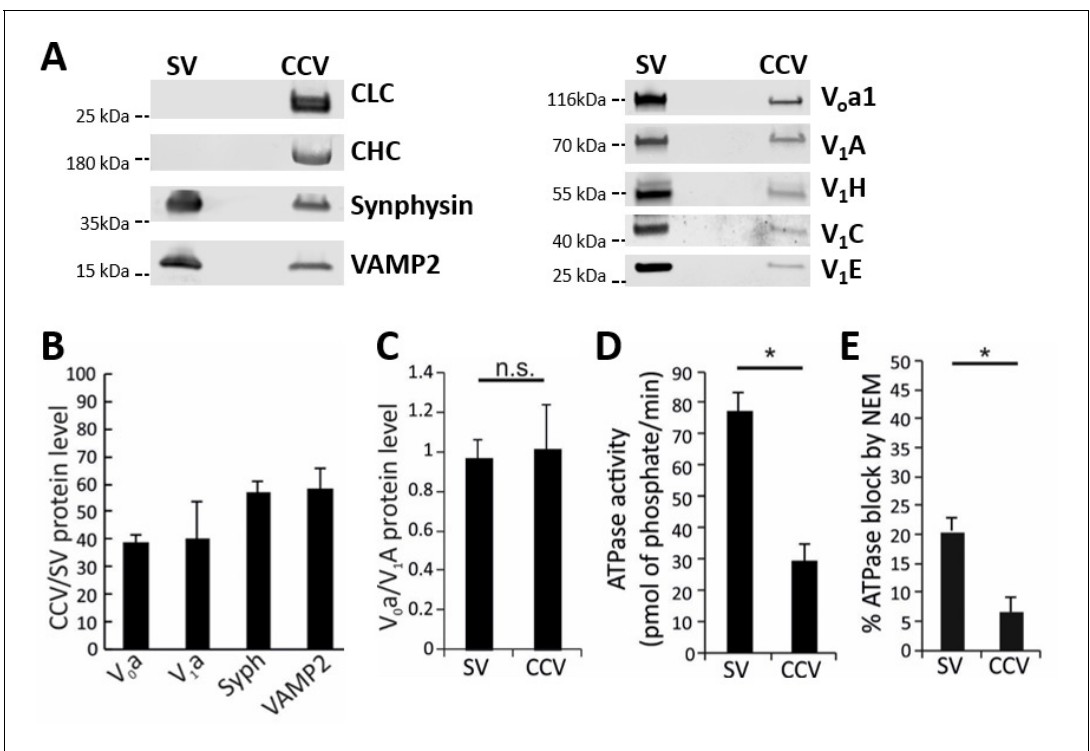

**Figure 3.** Functional analysis of the vATPase on CCVs. (**A**) Immunoblots of isolated SVs and CCVs for the clathrin light (LC) and heavy chains (HC), and SV marker proteins synaptophysin (Syph) and VAMP2 (left panel), as well as various $V_o$ and $V_1$ subunits of vATPase (right panel). (**B**) The ratio of the $V_o$ and $V_1$ as well as synaptophysin and VAMP2 (as SV markers) detected by immunoblotting in equal protein amount of CCVs and SVs (**C**) Normalized levels of $V_o$ and $V_1$ in CCV and SV samples, indicating that $V_o$:$V_1$ ratio is 1:1 in both preparations. (**D**) ATPase activity measured in 1.3 μg of isolated SVs and CCVs. (**E**) Blocking percentage of ATPase activity by NEM (vATPase inhibitor) in 1.3 μg of SVs and CCVs. Error bars in (**B–E**) represent SD of 3–4 experimental replicates (p<0.01 for D and E, and >0.05 for C).

DOI: https://doi.org/10.7554/eLife.32569.011

The following figure supplements are available for figure 3:

**Figure supplement 1.** Both $V_o$ and $V_1$ subunits are present on CCVs.

DOI: https://doi.org/10.7554/eLife.32569.012

**Figure supplement 2.** ATPase activity measurements in isolated SVs.

DOI: https://doi.org/10.7554/eLife.32569.013

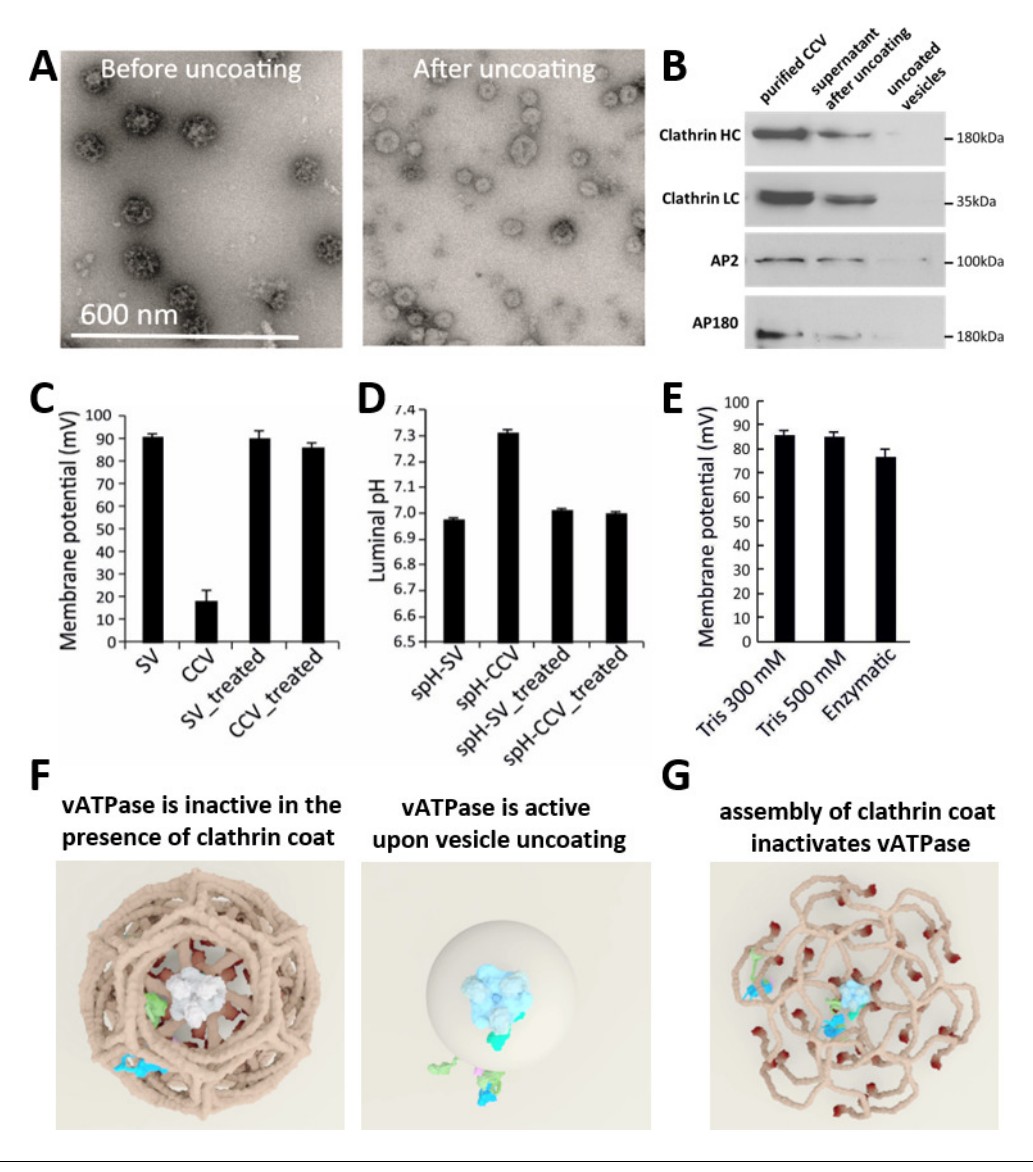

**Figure 4.** CCV uncoating revealed that the vATPase is blocked by clathrin coat. (**A**) Electron micrographs of negatively stained CCVs before and after uncoating with 300 mM Tris-buffer pH 9.0. (**B**) Western blot of CCVs, uncoated vesicles and supernatant after uncoating shows dissociation of clathrin LC and HC, as well as AP180 and AP2, from the uncoated vesicle (proteins were separated at the 10% gel and detected by chemiluminescence). (**C–D**) Membrane potential (**C**) and luminal pH (**D**) of acidified SVs and CCVs before and after treatment with Tris-buffer (pH 9.0). Error bars represent SD of 3–4 experimental replicates done on over 1000 vesicles each. (**E**) Membrane potential of acidified CCVs after treatment with 300 mM Tris-buffer (pH 9.0), 500 mM Tris buffer (pH 7.0) and 'enzymatic' treatment with 1.7 µg auxilin, 4.8 µg Hsc70 and 1.7 µg Hsp110 proteins. Error bars represent SD of 3 experimental replicates done on over 1000 vesicles each. (**F–G**) Model of vATPase block by clathin coat: solved structures of vATPase, clathrin tripods and AP2 complex were used to check how vATPase fits within the clathrin lattice. The plasma membrane is depicted in light beige, clathrin triskelia in dark beige/brown; vATPase complex in gray (when inactive), light blue (when active) and dark green ($V_1H$-subunit); AP2 complex in purple/blue/light green. As clathrin triskelia are recruited (through AP2), clathrin ring starts building around the vATPase complex. Insertion of the last triskelion of the clathrin ring would collide with the regulatory $V_1H$-subunit of vATPase (**G**), thus we hypothesize that the displacement of regulatory $V_1H$-subunit inwards results in the block of the vATPase activity. For more details, see Suppl. Data.
DOI: https://doi.org/10.7554/eLife.32569.014

The following figure supplements are available for figure 4:

**Figure supplement 1.** CCV uncoating revealed that the vATPase is blocked by clathrin coat.

*Figure 4 continued on next page*

*Figure 4 continued*

DOI: https://doi.org/10.7554/eLife.32569.015

**Figure supplement 2.** Proposed model of vATPase block by clathin coat.

DOI: https://doi.org/10.7554/eLife.32569.016

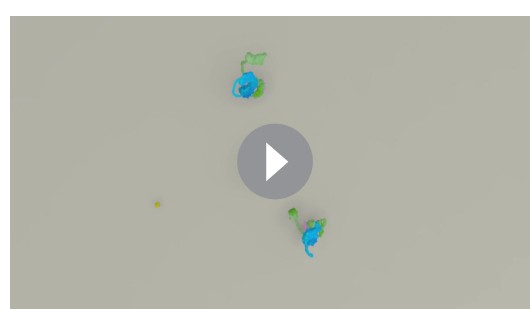

**Video 2.** Animated 3D model of vATPase block by clathin coat (top view). Note that all animations show the same process seen from different camera positions. Solved structures of vATPase, AP2 complex and clathrin tripods were used to see how vATPase fits within the clathrin lattice, thus all proteins have their 'true' dimensions. The plasma membrane is depicted in light beige, clathrin triskelia in dark beige/brown; vATPase complex in gray (when inactive), light blue/gray/dark green (when active; dark green = $V_1H$-subunit), AP2 complex in purple/blue/light green. The plasma membrane starts as a flat surface on which the clathrin triskelia begin forming the ring formation around the vATPase (in clockwise direction). Only two AP2 complexes are depicted: upon cargo (yellow) binding, the AP2 complex alters its structure and recruits clathrin triskelia that start the formation of a clathrin ring around the vATPase. For the clathrin ring to be 'closed', the 'last' triskelion to be inserted into the ring collides (gets in direct contact) with the vATPase, namely its regulatory $V_1H$-subunit. We hypothesize that the regulatory $V_1H$-subunit needs to be displaced inwards, resulting in the inhibition of the stalk rotation and thus block of the vATPase activity. So, upon insertion of the last triskelion in the clathrin ring, the vATPase becomes inactive (shown by a loss of color). The mechanical work of the clathrin coat proteins results in the membrane being pulled in, and as vesicle formation progresses, more clathrin triskelia are added. After vesicle is endocytosed, the clathrin coat disassembles (i.e. clathrin and adaptor proteins dissociate from the newly-formed vesicle), and the vATPase becomes active again. For more details, see Construction of the animated 3D model in the Supplementary Methods. A detailed (zoomed) view of regulatory $V_1H$-subunit displacement is shown in *Video 4*.

DOI: https://doi.org/10.7554/eLife.32569.017

## Discussion

Since, according to our knowledge, there is no evidence of a specific protein-protein interaction that blocks the vATPase activity at the newly formed vesicles at the synapse in the presence of clathrin or adaptor proteins, we propose that the vATPase may be sterically hindered by the three-dimensional clathrin scaffold that includes both clathrin triskelia and adaptor proteins (e.g. AP2, AP180, etc., adaptor proteins are essential to recruit clathrin triskelia to the membrane in order to initiate and organize the formation of clathrin coats). To see how the vATPase fits within the clathrin coat, we used the information provided by our structural analysis (*Figure 1D*, *Figure 1—figure supplement 2* and Suppl. Data), as well as previously solved structures of the vATPase (*Zhao et al., 2015*; *Mazhab-Jafari et al., 2016*) and the clathrin coat (*Fotin et al., 2004*). Considering the size of vAT-Pase and its $V_1$ domain (~25 nm and ~16 nm, respectively; *Oot et al. (2016)*; *Zhao et al. (2015)*, *Mazhab-Jafari et al., 2016*), we docked the solved structures of vATPase (*Zhao et al., 2015*) into a hexagonal ring of a clathrin coat (vATPase does not fit into a pentagonal ring) (*Figure 4F–G*; *Figure 4—figure supplement 2*; *Videos 2–4*, for more information see Suppl. Data). Previously published studies, as well as our cryo-EM analysis of the clathrin coat, have revealed that upon complete hexagonal ring

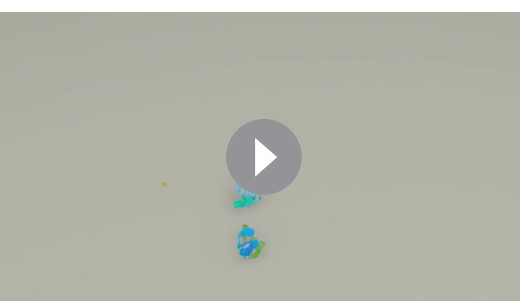

**Video 3.** Animated 3D model of vATPase block by clathin coat (side view). Note that all animations show the same process seen from different camera positions). For details, see legend to *Video 2*.

DOI: https://doi.org/10.7554/eLife.32569.018

formation, the terminals of clathrin HCs are placed in a very close proximity (<5 nm) to the vesicle membrane (*Cheng et al., 2007*). These terminal domains are suggested to form barrels with diameters ~ 10 nm at the inside surface of the cage (our data and *Fotin et al., 2004*). To fit in, the neck region of the vATPase (~13 nm) where the regulatory $V_1H$-subunit of the catalytic $V_1$ domain is located would need to be constricted (for more information, see *Figure 4—figure supplement 2* and Suppl data). Thus, we propose a hypothetical model where the steric hindrance provided by the formation of the clathrin coat around the vesicles blocks the vATPase activity (*Figure 4F*, *Figure 4—figure supplement 2*, *Videos 2–4*). Such a model is consistent with numerous publications (e.g., *Ho et al., 1993*; *Parra et al., 2000*; *Jefferies and Forgac, 2008*; *Diab et al., 2009*; *Oot and Wilkens, 2012*; *Oot et al., 2016*; *Zhao et al., 2015*; *Suzuki et al., 2016*; *Mazhab-Jafari et al., 2016*) and all our data, including the significantly reduced vATPase activity that we observed in isolated CCVs compared to SVs. It also allows for multiple layers of clathrin-coat formation regulation and provides a simple, yet elegant way by which the presence of a clathrin coat on the vesicle surface blocks the acidification process during its formation.

Our finding that clathrin coat-mediated vATPase inhibition prevents CCV acidification also resolves a longstanding debate in the field. It contributes to better understanding of the discrete steps of the SV cycle, now requiring clathrin uncoating upstream of acidification and neurotransmitter refilling. Interestingly, our model suggests that even partial formation of the clathrin coat around the vATPase may be sufficient to block its activity - such regulation may conserve ATP at the synapse. Finally, our finding that CCVs isolated from brain do not acidify matches findings in liver CCVs (*Fuchs et al., 1994*), and may provide insight into other cellular processes that involve both clathrin and vATPases.

## Materials and methods

### Animals

spH21 (*Li et al., 2005*) mice were provided by Dr. V. N. Murthy (Harvard University, USA) and Dr. W. Tyler (Virginia Tech Carilion Research Institute, USA). Wild-type mice were obtained from the animal facility of Max-Planck Institute for Biophysical Chemistry (MPIbpc), Göttingen, Germany, or purchased from Janvier.

### Isolation of synaptic vesicles and clathrin coated vesicles

SVs were isolated from brains of mice as described by *Ahmed et al. (2013)* and *Farsi et al. (2016)*. To isolate clathrin-coated vesicles (CCVs) from transgenic and wild-type mouse brains, we adapted a protocol from *Maycox et al. (1992)*. Fifteen wild-type or transgenic mice (6–8 weeks postnatal) were decapitated and their brains were homogenized in ice-cold Buffer-A (MES 100 mM pH 6.5, EGTA 1 mM, MgCl$_2$0.5 mM) supplemented with 200 µM PMSF (phenymethylsulfonyl fluoride) and 1 µg/ml pepstatin-A. The homogenate (H) was centrifuged (20,000 *g*, 20 min) and the resulting supernatant (S1) was further centrifuged at 55,000 *g* for 1 hr. The obtained pellet (P2) was re-suspended in Buffer A and diluted in Buffer A containing sucrose and Ficoll at final concentration of 6.25% wt/vol followed by centrifugation at 40,000 *g* for 40 min. The supernatant (S3) was then diluted 1:5 in Buffer A and subjected to ultracentrifugation (100,000 *g*, 1 hr). The obtained pellet (P4) contained coated vesicles, as confirmed by EM and immunoblotting. The pellet (P4) was next re-suspended in Buffer A and subjected to further centrifugation (20,000 *g*, 20 min). The obtained supernatant (S5) containing CCVs as well as other portion of SVs was overlaid on top of buffer A prepared with D$_2$O containing 8% (wt/vol) sucrose and centrifuged at 25,600 rpm for 2 hr in a Beckman SW28 rotor. The final pellet containing pure clathrin-coated vesicles was resuspended in sucrose buffer (sucrose 320 mM, HEPES 10 mM, pH 6.5). Characterization of purifed SV and CCV samples was performed using western blotting and antibodies listed in *Table 1*, as well as mass spectrometry.

### Immobilization of SVs and CCVs on glass coverslips

Glass coverslips were cleaned by sonication in 2% Hellmanex-III solution (Hellma Analytics), and coated with 0.1% (wt/vol) poly-L-lysine (PLL) before use. 50–100 ng of SVs or 0.5–1 µg of CCVs were diluted in the assay buffer (glycine 300 mM, MOPS 10 mM, pH 7.4) and immobilized on the

coverslips for 1 hr. The coverslips were then washed to remove the non-adsorbed vesicles and mounted in custom-made imaging chambers.

## Measurement of ΔpH and Δχ in single SVs and CCVs

Single-vesicle imaging was performed as described previously (*Farsi et al., 2016*). We observed no difference between ratiometric values of spH in purified vesicles before and after addition of carbonyl cyanide-4-(trifluoromethoxy)phenylhydrazon (FCCP) at pH 7.4, while acidifying the lumen by incubation of vesicles at pH 5.5 in the presence of FCCP significantly decreased the ratiometric value. This indicates that vesicles lose their luminal proton contents and reach equilibrium with their surrounding buffer during purification of vesicles and immobilization on glass coverslips. Acidification measurements were performed with vesicles isolated from the brain of transgenic mice (spH-vesicles). Potentiometric measurements were performed with the vesicles isolated from brains of wild-type animals after labeling with 100 nM of VF2.1.Cl. Acidification and potentiometric assays were performed in glycine buffer (300 mM glycine, 4 mM $MgSO_4$, 10 mM MOPS, pH 7.4) which was free of membrane-permeable ions. Thus, it could be assumed that the contribution of ions other than protons to the acidification was negligible. Moreover, it can be assumed that these buffer conditions vATPase was the main protein responsible for the observed difference between SVs and CCVs. For both types of measurements, immobilized vesicles were imaged using a total internal reflection fluorescence (TIRF) microscope (*Farsi et al., 2016*). The samples were excited with the 488 nm line of an argon laser and the emission was collected through a 515/30 nm filter. Images were acquired using the Andor IQ2 software. The fluorescence changes induced after addition of the same concentration of $Mg^{2+}$-ATP to SVs and CCVs were collected over time and corrected for bleaching of the fluorophore over the same experimental timescale before conversion to pH and membrane potential.

## Image analysis

The time series images were loaded as 3D stacks in MATLAB (Mathworks, Natrick, MA) and spot detection was performed using a previously described script (*Olivo-Marin, 2002*). A size cutoff (<20 pixels), as well as an eccentricity cutoff (<0.8 defining one as a line), were applied to the detected spots in order to remove aggregated particles from the analysis. The background for each vesicle was defined locally as the average intensity of neighboring pixels with the lowest intensity, and subtracted from the intensity of the spot in each frame. The fluorescence intensities were normalized to the integrated intensity of vesicles before the chemical perturbations. The normalized intensity, $F_{norm}(t)$ at time t, was then converted to pH and membrane potential using *Equation 1 and 2*, respectively.

$$\text{pH}_{(t)} = \text{pK}_a - \log_{10}\left[\frac{\left(1 + 10^{\text{pK}_a - 7.4}\right) F_{norm}(t)}{F_{norm}(t)}\right] \tag{1}$$

where the pKa is equal 7.2 (*Farsi et al., 2016*).

$$\Delta\Psi = k_{VF2.1.Cl} \, x \left(\frac{F(t) - F_0}{F_0}\right) \tag{2}$$

where $k_{VF2.1Cl}$ is equal 370.37 (*Farsi et al., 2016*).

## Statistical analysis

For comparisons between SVs and CCVs a two-sided Student's t-test for unpaired samples was used (p values indicated in the respective figures), unless otherwise indicated.

## Electron microscopy

Isolated SVs and CCVs were visualized by negative stain electron microscopy (EM). The main purpose of these experiments was to estimate the purity of the CCV sample (in addition to Western blotting and mass spectrometry), since the presence of contamination and uncoated SVs could interfere with the measurements. The EM detection of CCVs was also used to determine the size of CCVs, and the damage possibly inflicted on the coat by the purification procedure. Briefly, vesicles were applied to a formvar carbon-coated grid, washed with resuspension buffer, stained with 2%

**Table 1.** Antibodies used in this study.

| Antibody | Characteristics | Producer |
|---|---|---|
| ATP6V1A | NBP1-33021 Polyclonal | Novus-Biologicals |
| ATP6V1H | Ab187706 Polyclonal | Abcam |
| ATP6V1C1 | Ab87163 Polyclonal | Abcam |
| ATP6V1E1 | Ab111733 Polyclonal | Abcam |
| AP-2 | A2730 Monoclonal | Abcam |
| AP-180 | 155002 Polyclonal | Synaptic Systems (SySy) |
| Clathrin-heavy chain (HC) | Ab2731 Monoclonal | Abcam |
| Clathrin-light chain (LC) | Ab9884 Polyclonal | EMD-Millipore |
| EEA1 | PA1-063A Polyclonal | ThermoFischer Scientific |
| $Na^+/K^+$ ATPase | Ab7671 Monoclonal | Abcam |
| RPT4 (PMSF) | Ab14715 Monoclonal | Abcam |
| Proton ATPase (116 kDa subunit) | 109003 Polyclonal | Synaptic Systems (SySy) |
| Synaptotagmin-1 | 105 011 Monoclonal | Synaptic Systems (SySy) |
| Synaptophysin-1 | 101 011 Monoclonal | Synaptic Systems (SySy) |
| Synaptobrevin-2/VAMP-2 | 104 211 Monoclonal | Synaptic Systems (SySy) |
| anti-mouse IgG (IR680) | P/N 925–68070 | LI-COR |
| anti-rabbit IgG (IR800) | P/N 926–32211 | LI-COR |
| anti-mouse IgG (H + L) HRP | 62–6520 | ThermoFischer Scientific |
| anti-rabbit IgG (H + L) HRP | 65–6120 | ThermoFischer Scientific |

DOI: https://doi.org/10.7554/eLife.32569.020

uranyl acetate for 1 min at room temperature, washed in distilled $H_2O$ and dried. Images were obtained at various magnifications by JEM 1011 electron microscope (80kV, JEOL, Germany), or by CM120 Philips electron microscope equipped with a TemCam 224A slow scan CCD camera (TVIPS, Gaunting, Germany). To measure the diameter of the CCVs, the obtained images were analyzed by Digital Micrograph 3.4 software (Gatan Inc.) (*Schuette et al., 2004*). The longest and shortest diameter of each vesicle was measured, and an average was calculated in order to determine the mean CCV diameter, which is then plotted as shown in *Figure 1—figure supplement 1*.

To determine the ability of primary labeled anti-clathrin heavy chain (CHC) antibody X22 (Abcam) to detect the clathrin coat in its native-coat conformation, the purified CCV sample was subjected to immuno-gold labeling and inspected by EM. In short, the purified vesicles were added to coated carbon grids and fixing by 1% formaldehyde, followed by quenching with 20 mM glycine buffer and subsequent immunostaining with anti-clathrin HC antibody.

## Uncoating of clathrin-coated vesicles

To remove the clathrin coat from the CCVs, ~90 µg of isolated CCVs were diluted in 850 µl of pre-warmed uncoating buffer (Tris/Cl 300 mM, pH 9.0), and incubated at 37°C for 15 min, as described by *Maycox et al. (1992)*. Alternatively, we also used another approach to uncoat ~30 µg CCVs by incubating them with 300 µl of 500 mM Tris/Cl, pH 7.0, 5 mM DTT, 1 mM PMSF at 4°C for 4 hr (these conditions, sometimes during the overnight incubation, are commonly used to extract coat proteins including clathrin, adaptor proteins, auxilin and Hsc70 from purified coated vesicles for further purification; *Prasad and Keen, 1991*). Lastly, auxilin, Hsc70 and Hsp110 proteins were expressed and purified as described previously (*Schuermann et al., 2008*). Purified auxilin (1.7 µg), Hsc70 (4.8 µg) and Hsp110 (1.7 µg) proteins were added to 2 µg of isolated CCVs in the presence of 1.2 mM $Mg^{2+}$-ATP in 100 mM MES pH 7.0, 20 mM imidazole, 25 mM KCl, 10 mM $(NH_4)2SO_4$, 2 mM Mg-acetate, 2 mM DTT (as in *Morgan et al., 2013*), for 15 min at room temperature.

For single-vesicle imaging of CCVs after uncoating, the vesicles were first immobilized on the PLL-coated coverslips as described above and were incubated in uncoating buffer for 15 min at 37°C before imaging. After uncoating, the coverslips were washed with the assay buffer and incubated in

this buffer for 10 min to remove the residual Tris. In order to make sure that this treatment does not interfere with the v-ATPase activity, the same experiments were performed with SVs.

For electron microscopy, uncoated sample (irrespectively of how unocating is achieved) was centrifuged at 120,000 $g$ for 15 min to separate the vesicles from the uncoated proteins. The resulting pellet was resuspended in sucrose buffer and used at appropriate dilution for EM.

For Western blot experiments, ~50 µg of isolated CCVs were diluted in 300 µl of pre-warmed uncoating buffer (Tris/Cl 300 mM, pH 9.0), and incubated at 37°C for 15 min. The sample was then centrifuged at 120,000 $g$ for 15 min to separate the vesicles from the uncoated proteins. The supernatant was then removed and concentrated roughly 3x using centrifugal vacuum concentrator (Concentrator 5301, Eppendorf), while the resulting pellet was resuspended in 80 µl SDS-PAGE buffer. Isolated CCVs (diluted 1:10), uncoated vesicles and supernatant after uncoating were then analyzed by Western blot.

## ATPase activity assay

The ATPase activity of 1.3 µg of isolated SVs and CCVs was measured in 96-well plate using the 'ATPase/GTPase Activity Assay Kit' (Sigma) according to the manufacturer's instruction. The acidification was performed by incubating the samples with 2 mM $Mg^{2+}$-ATP in the assay buffer (300 mM glycine, 10 mM MOPS pH 7.4) for 15 min at room temperature. In this assay, the phosphate released by vATPase upon ATP hydrolysis is measured. To test for the free phosphate content in the SV and CCV samples, control samples with the same amount of isolated vesicles in the absence of ATP were measured. The absorbance of controls was then subtracted from the samples before calculating the amount of released phosphate.

For measuring the effect of N-ethylmaleimide (NEM) on the measured ATPase activity, the samples were incubated with 1 mM NEM for 10 min at room temperature before addition of $Mg^{2+}$-ATP.

## Cryo-electron microscopy and image processing

For sample preparation, 200 mesh Quantifoil R2/2 copper grids were glow-discharged and 0.65 mg/ mL of the CCV preparation in 100 mM MES, 1 mM EGTA, 0.5 mM $MgCl_2$ was added to the grid. The sample was plunge frozen with an FEI Vitrobot Mark V in liquid ethane.

The data were collected on a Talos Arctica (FEI) at 200 kV with a field emission gun and a nominal magnification of 45,000 at the Cryo-EM Swedish National Facility. The images were collected with a Falcon II and a dose rate of 9.53 $e^-$ per $Å^2$ per second. The recording time was 2.5 s, resulting in 30 frames and a total dose of 23.8 $e^-$ per $Å^2$. The frames were motion corrected and dose weighted with MotionCorr (*Li et al., 2013*) and the CTF was estimated with Gctf (*Zhang, 2016*). The pixel size for the images was 3.25 Å/pixel. For the processing, images with a box size of 130Åx130Å were manually extracted with RELION (*Scheres, 2012*). A total of 43,711 manually-picked individual particles from 2836 images were used for the 2D classification through RELION 2.0. After the classification, a new stack was formed for one vesicle based on the vesicle size. The image stack-classified into 16 classes with no (C1) symmetry imposed. The particles from the most populated well-defined 3D class were combined into a final stack. This stack was re-classified in three classes with D6 symmetry enforced. The particles of the most populated class were refined within RELION in D6. The gold standard FSC was calculated in RELION. Molecular graphics and analyses were performed with the UCSF Chimera package, a visualization system for exploratory research and analysis, developed by the Resource for Biocomputing, Visualization, and Informatics at the University of California, San Francisco (supported by NIGMS P41-GM103311) (*Pettersen et al., 2004*). The structure has been deposited with the EMDB-ID #4335.

## Fitting vATPase structure into CCV structure and construction of the animated 3D model

To see how the vATPase fits within the clathrin coat, we used the information provided by our structural analysis (*Figure 1D*, *Figure 1—figure supplement 2*, *Figure 4—figure supplement 2*), as well as previously solved structure of the vATPase (*Zhao et al., 2015*; *Mazhab-Jafari et al., 2016*) and the clathrin coat (*Fotin et al., 2004*). Considering the size of vATPase and its $V_1$ domain (~25 nm and ~16 nm, respectively; *Oot et al. (2016)*; *Zhao et al. (2015)*, *Mazhab-Jafari et al., 2016*), we docked the solved structures of vATPase (*Zhao et al., 2015*) into a hexagonal ring of clathrin coat

(vATPase does not fit into a pentagonal ring)(*Figure 4E–F*; *Figure 4—figure supplement 2*; *Videos 2–4*, for more information see Suppl. Data). Previously published studies, as well as our cryo-EM analysis of the clathrin coat, have revealed that upon complete hexagonal ring formation, the terminals of clathrin HCs are placed in a very close proximity (<5 nm) of the vesicle membrane (*Cheng et al., 2007*). These terminal domains are suggested to form barrels with diameters ~ 10 nm at the inside surface of the cage (our data and *Fotin et al., 2004*). To fit in, the neck region of the vATPase (~13 nm) where the regulatory $V_1H$-subunit of the catalytic $V_1$ domain is located and protruded ~4 nm from the stalk would need to be constricted (based on *Zhao et al., 2015*; *Suzuki et al., 2016*; *Mazhab-Jafari et al., 2016*). It is known that the $V_1H$-subunit can inhibit the ATPase activity of eukaryotic vATPase when $V_1$ is dissociated from $V_o$ (*Parra et al., 2000*; *Diab et al., 2009*; *Jefferies and Forgac, 2008*). However, in holo ($V_oV_1$) vATPase the same subunit is required for the ATP hydrolysis (*Jefferies and Forgac, 2008*). Interestingly, recent biochemical and structural studies have revealed that the dual function of $V_1H$-subunit is dependent on its position relative to the other $V_1$ subunits (*Jefferies and Forgac, 2008*; *Oot et al., 2016*), and might involve interactions with these subunits (*Jefferies and Forgac, 2008*). Further, the inhibitory effect of the $V_1H$-subunit is thought to be a result of its rotation towards $V_1D$ (*Oot et al., 2016*) and $V_1F$ (*Jefferies and Forgac, 2008*), which compose the central rotor of the complex (*Marshansky et al., 2014*; *Zhao et al., 2015*; *Mazhab-Jafari et al., 2016*). Thus, we propose a model where the steric hindrance provided by the formation of the clathrin coat around the vesicles results in dislocation of the regulatory $V_1H$-subunit towards the stalk (*Figure 4G*; *Figure 4—figure supplement 2*, *Videos 2–4*). The displacement of the regulatory $V_1H$-subunit respectively closer to the other $V_1$ subunits, most probably $V_1D$ and $V_od$, and in turn would block the activity of the proton pump as suggested by several publications (*Diab et al., 2009*; *Jefferies and Forgac, 2008*; *Oot and Wilkens, 2012*; *Zhao et al., 2015*; *Suzuki et al., 2016*; *Mazhab-Jafari et al., 2016*). Such a model is hypothetical but consistent with numerous publications (*Ho et al., 1993*; *Parra et al., 2000*; *Jefferies and Forgac, 2008*; *Diab et al., 2009*; *Oot and Wilkens, 2012*; *Oot et al., 2016*; *Zhao et al., 2015*; *Suzuki et al., 2016*; *Mazhab-Jafari et al., 2016*) and all our data, including the significantly reduced vATPase activity that we observed in isolated CCVs compared to SVs. It also allows for multiple layers of clathrin-coat formation regulation and provides a simple, yet elegant way by which the presence of a clathrin coat on the vesicle surface blocks the acidification process during its formation.

The animated 3D model of the vATPase block by clathrin coat was constructed using the original structures (as detailed below) and the custom-written plug-ins (https://github.com/IraMilosevic/eLife-Farsi-Milosevic; copy archived at https://github.com/elifesciences-publications/eLife-Farsi-Milosevic) (*Milosevic, 2018b*) for the 3D software Autodesk Maya (Autodesk Inc., San Rafael, CA). Information on protein structures was obtained from PDB based coordinates and the Uniprot database (PDB codes: 1XI4, 5TJ5: complete 3J9T, 3J9U, 3J9V, autoinhibited 5BW9). References used to model clathrin coat, AP2 and vATPase are listed in the Reference section. All proteins have their 'true' dimensions. The α- and β-linkers of AP2 are constructed to enable them to reach their maximal length. vATPase states are defined according to the ~120

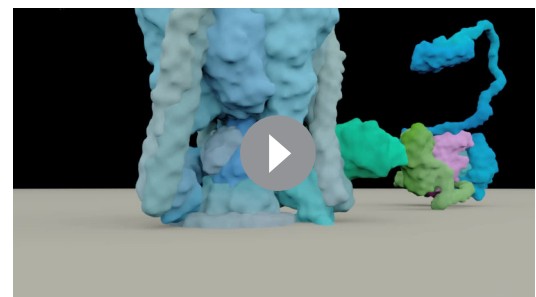

**Video 4.** Animated 3D model of vATPase block by clathin coat (front view). Note that all animations show the same process seen from different camera positions. Here, detailed view of vATPase at the level of plasma membrane is shown (overview of the endocytic vesicle formation is shown in the *Videos 2* and *3*). Note that the time duration of the breaks between the vATPase rotation cycles has been shortened to better illustrate the vATPase activity. Upon binding cargo protein, the nearby AP2 complex alters its structure and recruits clathrin triskelia. Clathrin ring starts to be built around the vATPase complex (only terminal domain, linker, ankle and a part of distal segment can be seen in this view): the insertion of the last clathrin triskelion in the ring displaces the $V_1H$-subunit of the vATPase. For more details, see Construction of the animated 3D model in the Supplementary Methods. After the vesicle is formed and endocytosed, the clathrin coat disassembles, and the vATPase subunits instantly resume 'original positions', bringing the vATPase in the active state again.
DOI: https://doi.org/10.7554/eLife.32569.019

°rotations: since autoinhibited $V_1$ is crystalized in state 2, and autoinhibited $V_o$ is crystalized in state 3, we chose state two to be the state where the stalk rotation is stopped when the clathrin N-terminus pushes it. We depict interplay of clathrin coat and vATPase in the context of a bare vesicle (40 nm in diameter): there are obviously many proteins on the vesicle, yet the proteins are likely mobile. Further, we detected all vATPase subunits on the CCVs, meaning that none of the subunit gets removed from the vATPase complex.

We referred to clathrin HC as 'HC'; $V_o$ subunits were referred to using lowercase letters i.e. 'd', and $V_1$ subunits were referred to using uppercase letters that is, 'D'; '$V_o$'/'$V_1$' lock refers to inactive vATPases; (structure) means that the protein/complex structure in this conformation is available. We considered five possible models:

(1)'HC' acts on 'd'/'D'/'F' -> '$V_o$'/'$V_1$' lock;

(2a) 'HC' acts on 'H' -> 'H' acts on 'D' (structure), possibly deforming/pushing it (structure) -> '$V_o$'/'$V_1$' lock;

(2b) 'HC' acts on 'H' -> 'H' acts on 'D' (structure), possibly deforming/pushing it (structure) -> '$V_1$' lock, 'd'/'D' interface disturbed -> '$V_o$' lock;

(3)'HC' acts on 'C' -> 'C' acts on 'd'/'D' -> '$V_o$'/'$V_1$' lock;

(4)'HC' acts only on 'a' -> 'a' acts on 'd' -> '$V_o$'/'$V_1$' lock;

(5)'HC' acts on unknown protein on the vesicle -> '$V_o$'/'$V_1$' lock.

While all of these models may be possible, based on our data and the published literature we consider that the model (2) is the most likely way how clathrin coat inhibits vATPases. Specifically, clathrin HCs that have their N-termini located directly around vATPase will have at least one N-terminus at which the structures collide at the level of the regulatory $V_1H$-subunit. Thus, $V_1H$-subunit should be displaced in order for the clathrin hexagonal ring to reach its final position. Notably, the cytosolic part of the vATPase (structure 5BW9) adopts an autoinhibited structure that shows the $V_1H$ subunit affecting the $V_1D$ subunit. We hypothesize that the formation of clathrin lattice leads to a similar structure in which the proton pump rotation is blocked. Such a model is animated in 3D, and presented in *Videos 2–4*. Note that all movies show the same process seen from different camera positions (top view *Video 2*, side view *Video 3*, front view *Video 4*).

## Acknowledgements

The cryo-EM data were collected at the Cryo-EM Swedish National Facility funded by the Knut and Alice Wallenberg Family, Erling Persson and Kempe Foundations, SciLifeLab, Stockholm University and Umeå University. The structure has been deposited with the EMDB-ID: 4335. We thank R Fernandez-Busnadiego (MPI for Biochemisty, München), D Riedel (EM Facility, MPI for Biophysical Chemistry, Göttingen) and W Möbius (EM Facility, MPI for Experimental Medicine, Göttingen) for their expert help and advice regarding the EM data, and J Rubinstein (Hospital for Sick Children, Toronto) for the discussions on the vATPase structure. We thank MPIbpc and ENI animal facilities, B Barg-Kues and M Costa for genotyping, K-W Li (University of Amsterdam) and CK Frese (Proteomics Core Facility, University of Cologne, CECAD Research Center) for the help with the MS, and N Raimundo for the help with proteomics. This work was supported by Schram-Stiftung T287/25457 and Deutsche Forschungsgemeinschaft (Emmy Noether Young Investigator Award MI-1702/1 and SFB889/A8) to IM, Human Frontier Science Program (Young Investigator Grant RGY0074/16) to CM, Peter-und-Traudl-Engelhorn fellowship to ZF, SySy fellowship to SG, and the NIH Grant GM118933 to EML. Conceptualization IM and RJ; Investigation and/or Analysis ZF, SG, MK, AW, CM, EML and IM; Model BR, SG and IM; Writing ZF and IM, all coauthors contributed to the final ms. The authors declare no competing financial interests.

## Additional information

### Competing interests

Reinhard Jahn: Reviewing editor, *eLife*. The other authors declare that no competing interests exist.

## Funding

| Funder | Grant reference number | Author |
| --- | --- | --- |
| Engelhorn Stiftung | Postdoc fellowship | Zohreh Farsi |
| Synaptic System | PhD fellowship | Sindhuja Gowrisankaran |
| National Institutes of Health | GM118933 | Eileen M Lafer |
| Human Frontier Science Program | Young Investigator Grant RGY0074/16 | Carsten Mim |
| Deutsche Forschungsgemeinschaft | Emmy Noether Young Investigator Award MI-1702/1 | Ira Milosevic |
| Schram Stiftung | T287/25457 | Ira Milosevic |
| Deutsche Forschungsgemeinschaft | SFB88/A8 | Ira Milosevic |

The funders had no role in study design, data collection and interpretation, or the decision to submit the work for publication.

## Author contributions

Zohreh Farsi, Data curation, Formal analysis, Supervision, Investigation, Methodology, Writing—original draft; Sindhuja Gowrisankaran, Data curation, Formal analysis, Validation, Investigation, Methodology, Writing—original draft; Matija Krunic, Data curation, Formal analysis, Investigation, Methodology; Burkhard Rammner, Resources, Visualization; Andrew Woehler, Resources, Formal analysis; Eileen M Lafer, Resources, Methodology; Carsten Mim, Resources, Data curation, Formal analysis, Investigation, Visualization; Reinhard Jahn, Conceptualization, Resources, Funding acquisition, Writing—original draft; Ira Milosevic, Conceptualization, Resources, Data curation, Formal analysis, Supervision, Funding acquisition, Investigation, Visualization, Methodology, Writing—original draft, Project administration

## Author ORCIDs

Reinhard Jahn http://orcid.org/0000-0003-1542-3498
Ira Milosevic http://orcid.org/0000-0001-6440-3763

## Ethics

Animal experimentation: Animal experiments were conducted according to the European Guidelines for animal welfare (2010/63/EU) with approval by the Lower Saxony Landesamt fur Verbraucherschutz und Lebensmittelsicherheit (LAVES), registration number 14/1701.

## Decision letter and Author response

Decision letter https://doi.org/10.7554/eLife.32569.028
Author response https://doi.org/10.7554/eLife.32569.029

# Additional files

## Supplementary files

• Supplementary file 1. Proteins identified from SV and CCV preparations by mass spectrometry. 20 µg protein from SV and CCV preparations were assayed by Q Exactive Mass Spectrometer (Thermo Scientific). MS/MS spectra were searched against an IPI mouse database (IPI, Mouse v3) with the ProteinPilot software (Applied Biosystems, USA). Common proteins from two trials are displayed. The identified vATPase (both $V_o$ and $V_1$) subunits are labelled in brown.
DOI: https://doi.org/10.7554/eLife.32569.021

• Transparent reporting form
DOI: https://doi.org/10.7554/eLife.32569.022

## Major datasets

The following dataset was generated:

| Author(s) | Year | Dataset title | Dataset URL | Database, license, and accessibility information |
|---|---|---|---|---|
| Farsi Z, Gowrisankaran S, Krunic M, Rammner B, Woehler A, Lafer EM, Mim C, Jahn R, Milosevic I | 2018 | Clathrin Coated Vesicle of 850 angstrom from mouse brain | http://www.ebi.ac.uk/pdbe/entry/emdb/EMD-4335 | Publicly available at the Electron Microscopy Data Bank (accession no. EMD-4335) |

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
