## [Decision Letter]

Thank you for submitting your article "Clathrin coat controls synaptic vesicle acidification by blocking vacuolar ATPase activity" for consideration by *eLife*. Your article has been favorably evaluated by Randy Schekman (Senior Editor) and three reviewers, one of whom is a member of our Board of Reviewing Editors. The reviewers have opted to remain anonymous.

The reviewers have discussed the reviews with one another and the Reviewing Editor has drafted this decision to help you prepare a revised submission.

Summary:

This study investigates the role of the clathrin coat in regulating the vATPase activity of the underlying vesicle. The authors made use of a transgenic mouse expressing synaptopHluorin (spH), a chimeric construct consisting of VAMP2 followed by the pH-sensitive GFP derivative pHluorin. They then used well-established methods to prepare either clathrin-coated vesicle (CCV)-enriched fractions or synaptic vesicle (SV)-enriched fractions from the brains of both the transgenic mice and wild-type mice. The vesicles were immobilised onto coverslips, ATP was added, and TIRF microscopy was used to measure either the change in pH, using vesicles from the spH-expressing mice, or the change in membrane potential, using vesicles from the wt mice and a potentiometric probe. In both cases, the ATP-induced changes were much greater in the SVs than in the CCVs. However, they found that if they first uncoated the CCVs by treating them with a pH 9.0 buffer, the uncoated vesicles displayed the same ATP-dependent changes in lumenal pH and membrane potential as the SVs. These data strongly indicate that the coat inhibits the vesicle's vATPase activity.

The reviewers all found the work interesting, particularly the way the authors used the spH transgenic mouse to carry out studies on individual CCVs and SVs. However, they also all agreed that the model proposed by the authors is very speculative, that the assay used for uncoating the CCVs is far from physiological, and that some additional experiments are needed.

Essential revisions:

1) The authors mention in their cover letter that they had tried uncoating the CCVs using similar conditions to those that are used in the cell, i.e. via auxilin and Hsc70, and didn't get full uncoating. However, it would be good to see the actual data. How incomplete was the uncoating? Did the authors see at least a partial increase in vesicle acidification? All three reviewers wanted to see data on vesicle uncoating under more physiological conditions, so it is likely that the readers will too.

2) Another key experiment, suggested by one of the reviewers and supported by the others, is to add back coat components to the uncoated vesicles. This would both help to establish that the uncoating method they used is valid, and also allow the authors to look at the role of individual coat proteins (i.e., not only clathrin but also AP-2, AP180, etc.).

3) The model needs to be toned down because there is really no experimental evidence to support it. If they can show, by carrying out the experiment suggested above, that it is clathrin and only clathrin that is inhibiting the vATPase, then their model has some support, but the method they used for uncoating clearly removes other proteins as well.

4) There are some problems with the immunoblots.

4a) First, there is no information anywhere about what antibodies were used. This is essential information because there is so much variability in the quality of individual antibodies. Was the same anti-clathrin heavy chain antibody used throughout? In Figure 1C, lots of non-specific bands seem to be appearing, while in Supplementary Figure 8, it gives the sort of strong and specific signal one would expect, especially in purified CCVs.

4b) The dot blots (Figure 4B and Figure 1—figure supplement 1) are not acceptable. The dots are of varying sizes and appearances, and the method assumes that all of the antibodies are entirely monospecific, when clearly some of them are not. They need to be replaced by Western blots.

4c) There are some inconsistencies in the Western blots. In particular, in the blot in Figure 3A, where equal amounts of protein were loaded into the SV and CCV lanes, there is a stronger signal for VAMP2 and other SV proteins in the SV lane than in the CCV lane, which is understandable because much of the protein in the CCV lane is clathrin and other coat proteins. Similarly, in Supplementary Figure 8, there is a stronger signal for VAMP2 in the SV lane. However, there is also a stronger signal for α-adaptin (AP-2) in the SV lane, which doesn't make sense. Moreover, the iBAQ values, which are reasonably quantitative within a single sample, clearly indicate that this is not the case. If one normalizes the iBAQ data to assume that there is twice as much VAMP2 in the SVs as in the CCVs (which is the sort of difference suggested by the Western blot), then there should be about four times as much α-adaptin in the CCVs as in the SVs, not less α-adaptin in the CCVs.

5) The authors say: "Clathrin-mediated endocytosis is a classic example of vesicle formation mediated by a coat assembly, and it is the prominent endocytic pathway at the synapse." This sentence should be toned down as CME is not necessarily the sole endocytosis pathway for synaptic vesicle endocytosis (Watanabe et al. 2013; Kononenko et al. 2014). Moreover, it is very relevant to this study to know where clathrin is used. If it is on the plasma membrane, then acidification will take place in the CCV or, as proposed here, after uncoating. However, if clathrin is required on endosome-like vesicles (Watanabe et al., 2014) then acidification could occur before clathrin coated bud formation, making the issue of acidification less relevant.

6) The difference in CCVs vs. SVs is revealed by assays performed in a buffer composed of 300 mM glycine, 4 mM MgSO4, 10 mM MOPS, pH 7.4, devoid of "membrane permeable ions". How is the pH equilibrated to 7.4? In this buffer, SVs can acidify only to pH 7.0, far from the physiological pH. In the "full characterization" of SVs, Figure 2—figure supplement 2, by adding 30 mM chloride SVs can acidify to pH 5.5, the physiological pH. Why did the authors use the buffer without membrane permeable ions? They should repeat the measures with chloride to assess whether CCVs are able to acidify or not in this more physiological context.

7) Is there any evidence that acidification is delayed at synaptic terminals? Available data on acidification, based on spH measurements in cultured neurons, seems to be at odds with a significant delay (Atluri and Ryan 2006; Leitz and Kavalali 2011; Soykan et al., 2017). This should be discussed. Moreover, there are mutant mice (synaptojanin KO, endophilin TKO) for which CCV uncoating seems to be delayed. Is there any indication that acidification is delayed in these mice?

---

## [Author Response]

Essential revisions:1) The authors mention in their cover letter that they had tried uncoating the CCVs using similar conditions to those that are used in the cell, i.e. via auxilin and Hsc70, and didn't get full uncoating. However, it would be good to see the actual data. How incomplete was the uncoating? Did the authors see at least a partial increase in vesicle acidification? All three reviewers wanted to see data on vesicle uncoating under more physiological conditions, so it is likely that the readers will too.

This point has been well taken, and we have now carried out uncoating experiments using two additional and independent procedures. In the previous version, we used 300mM Tris-buffer pH 9 for 15 min to uncoat CCVs, as described by Maycox et al. JCB 1992. We have now also uncoated CCVs while maintaining neutral pH by treating them with the buffer containing 500mM Tris, pH 7.0, 5 mM DTT, 1 mM PMSF at 4C for 4 hours ° – these conditions are commonly used to extract coat proteins including clathrin, adaptor proteins and/or auxilin from purified coated vesicles for further purification (e.g., Prasad and Keen, Biochemistry 1991). Full uncoating of CCVs and recovery of full acidification was observed (see new Figure 4 in the revised manuscript).

Second, as requested by the reviewers, we have carried out uncoating experiments using purified auxilin, Hsc70 and Hsp110 proteins (through collaboration with Prof Eileen Lafer, San Antonio, TX, USA). We were pleased to observe that the addition of highly purified auxilin, Hsc70 and Hsp110 for 15 min at room temperature has uncoated all CCVs efficiently, and revealed that uncoated vesicles acidify (see new Figure 4).

In sum, as presented in new Figure 4, uncoating enhances acidification of the CCV sample to the level of SV sample regardless of whether uncoating was achieved enzymatically, or by treatment with “non-physiological” uncoating buffers.

2) Another key experiment, suggested by one of the reviewers and supported by the others, is to add back coat components to the uncoated vesicles. This would both help to establish that the uncoating method they used is valid, and also allow the authors to look at the role of individual coat proteins (i.e., not only clathrin but also AP-2, AP180, etc.).

We agree that such an experiment would be highly attractive, however, while the formation of empty clathrin cages in vitro (without vesicle) is well established for over a decade, we are not aware of any reports showing that uncoated vesicles can be re-coated in vitro by adding coat components.

Nevertheless, we have tried several conditions to recoat recently uncoated vesicles. In one of the approaches, we first uncoated freshly prepared CCVs using the aforementioned 300 mM Tris-buffer approach, and then added recombinant clathrin HC and LC obtained from Prof Lafer. While we have detected partial recoating (clathin coats were detected on ~9% of vesicles by EM, no coated vesicles were detected in uncoated samples) and a shift towards lower membrane potential values was observed when averaging the whole vesicle population, the recoating was not very efficient under these conditions. These data are now included below as additional data for reviewers. We spent several months trying to address this point, but apparently this is far from trivial as suggested by the lack of published evidence showing that such recoating can be achieved in vitro.

Additional data for reviewers:

Recoating of uncoated clathrin-coated vesicles

To remove the clathrin coat from the CCVs, ~20 µg of isolated CCVs were diluted in 300 µl of pre-warmed uncoating buffer (Tris/Cl 300 mM, pH 9.0), and incubated at 37 °C for 15 min. The sample was then transferred to dialysis cassette (Slide-A-Lyser 2K, Thermo Scientific) and dialysed overnight at 4°C against MES 100 mM pH 7.0, imidazole 20 mM, KCl 25 mM, (NH_4)2_SO_4_ 10 mM, Mg-acetate 2mM, DTT 2mM. Additional 1h was allowed at room temperature before the sample was carefully taken out of the dialysis cassette. The vesicles were immobilized on the PLL-coated coverslips and labeled with VF2.1.Cl as detailed above. Alternatively, for the EM experiments, the sample taken out of the dialysis cassette was centrifuged at the table-top centrifuge at maximal speed for 20 minutes, and 3-5 µl taken from the bottom of the tube were put to the formvar carbon coated 100-mesh grids. The samples are the stained with 2% uranyl acetate for 1 min at RT, dipped in distilled H_2_O, dried and subsequently inspected by JEM 1011 microscope (80kV, JEOL, Germany).

**Author response image 1. respfig1:** Partial recoating of previously uncoated CCVs was not robust enough to produce a significant difference in acidification assay. (**A**) Electron micrographs of negatively stained sample after the vernight “recoating” treatment as detailed in Materials and methods. While coat formation sometimes occurs without the vesicle at its center, several vesicles surrounded by a new coat are clearly seen (indicated by a white arrow). Scale bar 200 nm. (**B**) Membrane potential of acidified vesicles after the overnight “recoating” treatment Error bars represent SD of 3 experimental replicates.

**Author response image 2. respfig2:** Cryo-electron tomography on partially uncoated CCVs reveals that clathrin coat-vATPase interactions exist. (**A**) Electron micrographs of partially uncoated CCVs (in tightly packed CCVs, vATPase could not be detected). (**B**) Zoomed image of two vesicles at different views, as indicated. Red arrowhead points presumably to vATPase, blue arrowheads to clathrin filaments. See cryo-electron tomography Video for a better view.

3) The model needs to be toned down because there is really no experimental evidence to support it. If they can show, by carrying out the experiment suggested above, that it is clathrin and only clathrin that is inhibiting the vATPase, then their model has some support, but the method they used for uncoating clearly removes other proteins as well.

There might be a misunderstanding here. Our title, Abstract and manuscript text state clearly that it is the complete clathrin coat that controls synaptic vesicle acidification, and not clathrin itself. The proposed model is based on the contribution of adaptor proteins since adaptor proteins (e.g. AP2, AP180) are essential for recruiting clathin triskelia to the membrane in order to initiate and organize the formation of clathrin coat around the vesicle. It is not possible to form clathrin coats around vesicles without adaptor proteins. vATPase is the largest protein complex on synaptic vesicles that is, due to its size, most likely positioned in one of the hexagon rings of the clathrin coat. Given that we find the whole vATPase complex on CCVs (Figure 3 and the proteomics data), and that the structure of vATPase and clathrin coat are known, we have examined, by using the original structures in their real dimensions, how clathrin coat can accommodate the large vATPase complex. While we are not yet able to detect vATPase at CCVs in the cryo-EM experiments, we have succeeded in generating a reconstruction of most abundant CCV moiety (Figure 1—figure supplement 2 and Figure 1D), and we have docked the published vATPase structure into this reconstruction (Figure 3—figure supplement 2). The collision between vATPase structure and our reconstructed CCV (identical to the clathrin lattice in Fotin et al. Nature 2004, albeit our structure contains the vesicle) occurs primarily at the level of V_1_H regulatory subunit, a part of vATPase suggested to regulate the vATPase activity by numerous reports.

In order to obtain additional experimental evidence for the model, we have performed cryo-electron tomography on purified CCVs, and on purified CCVs that were partially uncoated (vATPase is not well visible in the tightly packed clathrin-coated vesicles) with a help of Dr Ruben Fernandez-Busnadiego at the MPI for Biochemistry, Munich, Germany. We have detected signals reminiscent of clathrin “filaments” in the close proximity of the signals reminiscent of vATPases, and the measured dimensions were in agreement with our model (see Video 1 and Author response image 2), but more work is needed here to ensure that the observed signals correspond to clathrin and vATPase. Thus, we prefer not to report on the cryo-electron tomography data at this stage.

We have now added additional data and information in the revised manuscript, and would like to stress that we consider that the clathrin coat, and not clathrin or adaptor proteins alone, block the vATPase activity. We have prepared an animation using original structures and following numerous cited literature, and we have stated at several instances that this is just a hypothetical model, and that more experimental work is needed to understand how clathrin coat inhibits the vATPase activity while the vesicles are coated.

4) There are some problems with the immunoblots.4a) First, there is no information anywhere about what antibodies were used. This is essential information because there is so much variability in the quality of individual antibodies. Was the same anti-clathrin heavy chain antibody used throughout? In Figure 1C, lots of non-specific bands seem to be appearing, while in Supplementary Figure 8, it gives the sort of strong and specific signal one would expect, especially in purified CCVs.

We have now included a table that lists all antibodies and their sources (see new Table 2). In the original manuscript, several different antibodies against clathrin LC and HC were used. We have repeated all experiments with anti-clathrin HC and LC antibodies obtained from Abcam and Millipore, respectively (see new Figure 1C).

4b) The dot blots (Figure 4B and Figure 1—figure supplement 1) are not acceptable. The dots are of varying sizes and appearances, and the method assumes that all of the antibodies are entirely monospecific, when clearly some of them are not. They need to be replaced by Western blots.

Figure 1—figure supplement 1: Please note that the dot blots were used just to optimize the CCV purification protocol. Specifically, we have monitored the distribution of clathrin and VAMP2, an integral synaptic vesicle protein, in the D_2_O/sucrose gradient by dot blot, since there was not enough material to run the classical Western blot. We have chosen anti- VAMP2 (SySy) and anti-clathrin LC (Millipore) antibodies, which are affinity-purified antibodies with proven specificity (numerous publications and manufacturers’ web pages, we have verified the specificity of these antibodies ourselves as well). If the reviewers still consider these data not acceptable, we will remove them. However, we consider that it is useful to show a key protocol optimization step in more details.

Figure 4B: It was reported that incubation with the Tris-buffer removes clathrin coat proteins (Maycox et al., 1992), and similar “stripping” approaches were often used to extract coat proteins (e.g., clathrin, adaptor proteins, auxilin) from coated vesicles for further purification. We have verified this also in our experiments, and dot blots were replaced by Western blots (see new Figure 4E).

4c) There are some inconsistencies in the Western blots. In particular, in the blot in Figure 3A, where equal amounts of protein were loaded into the SV and CCV lanes, there is a stronger signal for VAMP2 and other SV proteins in the SV lane than in the CCV lane, which is understandable because much of the protein in the CCV lane is clathrin and other coat proteins. Similarly, in Supplementary Figure 8, there is a stronger signal for VAMP2 in the SV lane. However, there is also a stronger signal for α-adaptin (AP-2) in the SV lane, which doesn't make sense. Moreover, the iBAQ values, which are reasonably quantitative within a single sample, clearly indicate that this is not the case. If one normalizes the iBAQ data to assume that there is twice as much VAMP2 in the SVs as in the CCVs (which is the sort of difference suggested by the Western blot), then there should be about four times as much α-adaptin in the CCVs as in the SVs, not less α-adaptin in the CCVs.

We thank the reviewers for this comment. We have quantified the protein concentration in both SV and CCV samples, and have loaded the same amount of protein to be able to compare two samples. As correctly pointed out by the reviewers, much of the protein in the CCV sample is contributed by clathrins and other coat proteins, hence a difference in the overall level of integral SV proteins (e.g. VAMP2). We have repeated Western blots for α-adaptin, with the similar vexing result as originally reported. At present, we cannot explain this discrepancy (maybe it is a problem of the antibody). Given that this minor point is not relevant for the key message of the manuscript, we have removed the α-adaptin Western blot data from the revised version.

5) The authors say: "Clathrin-mediated endocytosis is a classic example of vesicle formation mediated by a coat assembly, and it is the prominent endocytic pathway at the synapse." This sentence should be toned down as CME is not necessarily the sole endocytosis pathway for synaptic vesicle endocytosis (Watanabe et al. 2013; Kononenko et al. 2014). Moreover, it is very relevant to this study to know where clathrin is used. If it is on the plasma membrane, then acidification will take place in the CCV or, as proposed here, after uncoating. However, if clathrin is required on endosome-like vesicles (Watanabe et al., 2014) then acidification could occur before clathrin coated bud formation, making the issue of acidification less relevant.

We agree and have toned down the quoted sentence.

6) The difference in CCVs vs. SVs is revealed by assays performed in a buffer composed of 300 mM glycine, 4 mM MgSO4, 10 mM MOPS, pH 7.4, devoid of "membrane permeable ions". How is the pH equilibrated to 7.4? In this buffer, SVs can acidify only to pH 7.0, far from the physiological pH. In the "full characterization" of SVs, Figure 2—figure supplement 2, by adding 30 mM chloride SVs can acidify to pH 5.5, the physiological pH. Why did the authors use the buffer without membrane permeable ions? They should repeat the measures with chloride to assess whether CCVs are able to acidify or not in this more physiological context.

We have used a buffer without membrane permeable ions (as originally reported in Farsi et al., Science 2016) to exclude the contribution of any ion/proton exchanger to the difference that was observed between SV and CCV samples. That said, we have repeated these experiments in the presence of chloride now. Note that a significant difference between SV and CCV samples is present also under these conditions (new Figure 2—figure supplement 3).

Regarding the question about luminal pH of vesicles being equilibrated to 7.4, we have shown in our previous work (Farsi et al., 2016) that we did not observe any difference between ratiometric values of pHluorin-probe in purified vesicles before and after addition of FCCP (a proton ionophore) at pH 7.4, while acidifying the lumen by incubation of vesicles at pH 5.5 in the presence of FCCP significantly decreased the ratiometric value. This indicates that vesicles equilibrate with their surrounding buffer during purification and subsequent immobilization on glass coverslips.

7) Is there any evidence that acidification is delayed at synaptic terminals? Available data on acidification, based on spH measurements in cultured neurons, seems to be at odds with a significant delay (Atluri and Ryan 2006; Leitz and Kavalali 2011; Soykan et al., 2017). This should be discussed. Moreover, there are mutant mice (synaptojanin KO, endophilin TKO) for which CCV uncoating seems to be delayed. Is there any indication that acidification is delayed in these mice?

Quenching of the pH-sensitive fluorescent protein pHluorin reflects the combined rate of endocytosis and re-acidification of SV proteins tagged with pHluorin (e.g., Miesenböck et al., 1998; Granseth et al., 2006; Atluri and Ryan 2006; Balaji and Ryan, 2007; Balaji et al., 2008; Leitz and Kavalali 2011). Studies with pHluorin-based probes are useful, but not straight-forward since these probes are affected both by endocytic and/or acidification processes, and such processes are stimulation-, model- and temperature-dependent. By using synaptopHluorin, Atluri and Ryan (2006) have suggested that SVs re-acidify within approximately 4-5s. By using vGLUT-pHluorin, Leitz and Kavalali (2011) found that majority of endocytic events happen within 4s after stimulation. Using the rapid application of quenching solutions to separate endocytosis from acidification, Soykan et al. (2017) have also detected two endocytic phases, rapid and slow, at 37°C. In addition, Balaji et al. (2008) tried to separate the components of endocytosis and acidification: they found that the endocytic time constant (measured by the vGLUT-pHluorin probe) was ~6s at 36°C, and ~14s at 25°C, while vesicle acidification took ~4s.

Recent morphological studies based on optogenetic stimulation followed by rapid freezing proposed the existence of “ultrafast endocytosis” that can recover synaptic vesicle membrane with 30-300 ms after stimulation through the formation of synaptic endosomes (Watanabe et al., 2013a, 2013b, 2014). These synaptic endosomes could be further resolved by clathrin-mediated endocytic process into coated vesicles as fast as ~3s after stimulation (Watanabe et al., 2014).

Our finding that clathrin coat blocks vesicle acidification until the coat is removed corroborates several of aforementioned studies, and it contributes to the better understanding of these processes. Yet, as apparent from this short overview, the kinetics of vesicle recycling at the synapses is still intensely studied, and the contributions of temperature and stimulation strength are presently being examined. In addition, the nature of intermediates that occur after stimulation is not well understood. This is a complex topic that requires a detailed technical context to be adequately addressed, however, if is considered absolutely necessary, we will include it.

The delayed post-stimulus recovery observed through pHluorin quenching in auxilin (Yim et al., 2010), synaptojanin-1 KO (Mani et al., Neuron 2007) and endophilin-A TKO (Milosevic et al., Neuron 2011) mutants suggests a delayed kinetics of endocytic process, a delay in the acidification of the newly formed vesicles, or both (note that both mutants accumulate CCVs and endosome-like structures). Notably, based on our data (Milosevic et al. Neuron 2011) as well as Mani et al. (2007) and Yim et al. (2010), the loss of endophilin, synaptojanin-1 or auxilin does not cause a block in uncoating, but a kinetic delay. Similarly, neurotransmission is impaired, but not abolished, and functional vesicles can eventually be regenerated from coated structures (Milosevic et al., 2011). This point is now mentioned in the manuscript.